# Taming Massive Activations and Preconditioning Weights: GSR-Guided Quantization for W4A4

## Abstract

Large language model inference is constrained by memory and latency. Uniform low-bit quantization would help, but recent evidence shows massive activations—rare, extremely large, and largely input-invariant per-token scalars—rather than generic channel-wise outliers. Methods that "smooth" activation outliers by migrating scale into weights are therefore less effective under this phenomenon. We address this by explicitly rotating activations and preconditioning weights so that both become easy to quantize.

We first identify that the **grid-to-standard-deviation ratio (GSR)**, $\rho_g^X = \frac{\Delta_g}{\text{std}(X_{t,\cdot})}$, is a useful proxy for quantization sensitivity, as it measures the relative coarseness of quantization steps compared to the intrinsic variability of activations. Building on this insight, we introduce **Flattened Rotation tSVD Quantization(FRTQ)**, a post-training quantization framework tailored for ultra-low-bit settings (e.g., W4A4).

For activations (per-token), FRTQ learns orthogonal rotations at function-invariant points to contract GSR and stabilize quantization. For weights (per-channel), FRTQ fits a rank-$r$ truncated-SVD component to capture dominant directions, quantizes the residual, and realizes the correction via a fused low-rank path. All rotations are folded into adjacent weights, with only a single lightweight on-the-fly rotation required at the FFN down-projection.

By explicitly minimizing GSR, FRTQ aligns its updates with quantization error reduction. The method is purely post-training, requires only a small calibration set, and avoids gradient-based tuning. Its alternating updates are simple, scalable, and kernel-friendly. Experiments across standard LLM backbones show that FRTQ consistently reduces GSR and improves W4A4 accuracy compared to smoothing-only or rotation-only baselines. On LLaMA-2 70B, FRTQ lowers $\rho$ of activation by 28.69% compared to DFrot, and improves KV4 zero-shot accuracy by 1.25, matching higher-bit baselines while incurring negligible runtime overhead.

## 1 Introduction

The community has made LLMs ubiquitous in tools, agents, and assistants, yet the inference budget—both memory and latency—still constrains what can be deployed at scale. Quantization is one of the most practical levers for reducing that budget and has attracted sustained attention across PTQ and QAT pipelines. A persistent challenge, however, is the presence of *massive points* in both activations and weights: a tiny fraction of values can dominate the dynamic range and propagate unusually large rounding error (Sun et al., 2024).

Two lines of progress illustrate a common theme: add *degrees of freedom (DoFs)* that can be tuned before discretization. On the activation side, inserting orthogonal (or Hadamard) rotations at invariance points reshapes hidden-state distributions and reliably unlocks W4A4 inference (Ashkboos et al., 2024; Liu et al., 2025; Xiang & Zhang, 2025). On the weight side, low-rank structure via SVD can absorb dominant directions, leaving a residual that is easier to quantize (Li et al., 2025). At their core, these methods expose redundant DoFs and search within them for parameters that suppress massive points and, in turn, reduce quantization error.

Uniform quantization yields non-smooth errors, so we optimize a scale-invariant surrogate that respects bit-width: the grid–std ratio (GSR). For a $b$-bit asymmetric quantizer with

$$\Delta = \frac{x_{\max} - x_{\min}}{2^b - 1}, \qquad \sigma^2 = \tfrac{1}{n}\sum_i (x_i - \mu)^2,$$

define $\mathrm{GSR} = \Delta/\sigma$. Scaling a row by any $c > 0$ scales both $\Delta$ and $\sigma$ equally, making GSR invariant to channel-wise scaling and a faithful proxy for quantization difficulty.

Let $A \in \mathbb{R}^{n \times d}$ be activations and $W \in \mathbb{R}^{d \times h}$ weights. Per-row/column ratios are

$$\rho_i^{\mathrm{act}} = \frac{\Delta(A_{i,:})}{\sigma(A_{i,:})}, \quad \rho_j^{\mathrm{w}} = \frac{\Delta(W_{:,j})}{\sigma(W_{:,j})}.$$

Smaller GSR implies finer grid relative to spread and thus lower expected rounding error.

**Our approach: Flattened Rotation tSVD Quantization (FRTQ).** FRTQ exploits the *different* degrees of freedom offered by rotations and low-rank factorisations to drive down the grid–std ratio for *both* activations and weights, thereby suppressing their individual quantization errors and—by extension—the network-wide error.

*Activation step.* The orthogonal matrix $R_\ell \in \mathrm{O}(d)$ is nudged by a single Cayley/Procrustes update so that the rotated activations $R_\ell A_\ell$ attain a smaller $\mathcal{P}_\ell^{\mathrm{act}}$, i.e. lower GSR.

*Weight step.* A tunable low-rank patch $\Delta_\ell = U_\ell V_\ell^\top$ ($\mathrm{rank} \leq k$) is fitted to $W_\ell$; the extra $k(m+n)$ parameters re-allocate spectral mass, shrinking $\mathcal{P}_\ell^{\mathrm{w}}$ while the INT4 residual remains kernel-friendly.

The calibration loop avoids full quantization-aware training: no gradients flow through quantizers, no labels are needed, and the network is never fine-tuned end-to-end. Instead, we iteratively lower the grid–std ratio (GSR) of *activations* with a single Cayley update to the rotation matrix, and of *weights* with one row-wise tSVD update. Each pass trims the dominant per-row maxima—either by redistributing activation energy or by absorbing extreme weight directions into a tiny high-precision branch—until further GSR drops become negligible. Finally, the learned rotations and low-rank patches are fused back into the weights, so inference runs on standard INT4 kernels plus a negligible full-precision slice, achieving near-QAT accuracy at a fraction of the cost.

**Contributions.** **(i)** We introduce the *grid–std ratio* (GSR), a scale-invariant proxy defined as the quantization grid $\Delta$ divided by the channel-wise standard deviation $\sigma$. Under mild assumptions, the relative quantization error scales linearly with GSR, making it a unified, bit-width–aware metric for both activations and weights.
**(ii)** We propose **FRTQ**, a post-training pipeline that alternates a single Procrustes rotation update on activations with a row-tuned SVD correction on weights to drive down GSR. Using only a small calibration set and lightweight updates, FRTQ achieves near-QAT accuracy in W4A4 regimes without end-to-end tuning or significant extra compute.

## 2 RELATED WORK

**Scale/activation smoothing.** Activation range is a core PTQ bottleneck; *massive activations* are rare, input-insensitive, bias-like directions that dominate error (Sun et al., 2024). **SmoothQuant** shifts difficulty from activations to weights via per-channel scaling, enabling accurate W8A8 with modest calibration (Xiao et al., 2022). Refinements improve stability and policy: **Outlier Suppression+** offers fast, robust scaling (Wei et al., 2023); **OmniQuant** adds learnable clipping and equivalent transforms for ultra–low-bit PTQ (W-only and W+A) (Shao et al., 2023). For W4A8, **QQQ** adopts adaptive smoothing (Zhang et al., 2024), and **QServe** introduces *SmoothAttention* for 4-bit KV cache (Lin et al., 2024). These methods chiefly re-balance difficulty between weights and activations while not fully eliminating extreme tails (Sun et al., 2024).

**Rotation-based activation quantization.** A complementary strategy *disperses* activation energy via orthogonal transforms with no inference-time overhead. **QuIP** increases incoherence between weights and the Hessian using randomized orthogonals (Chee et al., 2023); **QuIP#** accelerates this with randomized Hadamards (Tseng et al., 2024). **QuaRot** leverages rotational invariance to insert

Hadamard/orthogonal transforms at invariant sites, yielding outlier-free hidden states and stable 4-bit activations (incl. KV) (Ashkboos et al., 2024). Moving to learned rotations, **SpinQuant** and **OSTQuant** refine on the Stiefel manifold via Cayley updates (Liu et al., 2025; Hu et al., 2025; Li et al., 2020). **DFRot** explains Hadamard's advantage, reweights calibration toward *massive* tokens, and applies Procrustes-style refinement for long-tail robustness (Xiang & Zhang, 2025).

**Low-rank and hybrid corrections.** Weight-side structure can *absorb* difficult directions before quantization. **SVDQuant** splits weights into a high-precision low-rank branch plus a quantized residual (Li et al., 2025). Related lines include **LoRA** (Hu et al., 2021), compensation-style PTQ such as **ZeroQuant-V2** (Yao et al., 2023), and efficient tuning on quantized models via **QLoRA** (Dettmers et al., 2023). Though often motivated by memory/compression, these structural views complement smoothing/rotation: absorbing bias-like, high-energy directions reduces the incidence and impact of massive activations (Sun et al., 2024) in matmul-heavy Transformer blocks.

## 3 PRELIMINARIES

**Uniform quantization** For a real-valued vector $x \in \mathbb{R}^d$ we record its calibration bounds $x_{\min}$ and $x_{\max}$, set the scale

$$\Delta = \frac{x_{\max} - x_{\min}}{2^b - 1}, \qquad z = \left\lfloor \frac{x_{\min}}{\Delta} \right\rceil,$$

and encode each element as

$$\mathcal{Q}_{\Delta,z}(x) = \Big( \text{clip}\big( \lfloor x/\Delta \rceil - z, \ 0, \ 2^b - 1 \big) + z \Big) \Delta.$$

Because the grid $\Delta$ is fixed by the *largest* magnitude in the channel, the quantizer is acutely sensitive to rare but extremely large values: a single "massive" activation can inflate $x_{\max}$, coarsen the entire grid, and dominate the rounding error budget for all other entries.

**Fusible orthogonal rotations.** Consider a linear transformation $y = Wx$ inside a network. Inserting an orthogonal matrix $R$ before the multiply gives

$$y = W(R^\top R)x = (WR^\top)(Rx) = \widetilde{W}\,\widetilde{x},$$

with $\widetilde{W} = WR^\top$ and $\widetilde{x} = Rx$. Because $R^\top R = I$, the overall mapping is unchanged; the rotation can be merged into the weight tensor once during calibration. This fusion adds no extra FLOPs or memory at inference yet grants $d(d-1)/2$ continuous degrees of freedom—helpful to redistribute activation energy, tame outliers, and reduces quantization error.

**SVD-based low-rank absorption.** Any $W \in \mathbb{R}^{m \times n}$ factors as $W = U\Sigma V^\top$. We keep the top-$k$ slice $W_{\text{hi}} = U_k \Sigma_k V_k^\top$ in full precision and apply it to full-precision $x$; since $k \ll \min\{m,n\}$, this is a small matvec. The residual $R_k = W - W_{\text{hi}}$ has tighter dynamic range and is jointly quantized with $x$, so most MACs run in low precision. The split exposes $k(m+n)$ tunable DoF (columns of $U_k$, $V_k$, and $\Sigma_k$) to further reduce error without extra latency. Conceptually, the SVD branch adds a low-rank "detour" that captures as much spectral energy as possible in high precision, maximizing the unquantized fraction of information. It does not, however, optimize the residual's quantization: the step size for $R_k$ is still set by its largest entry, so additional mechanisms (e.g., smarter rank allocation or complementary rotations) are needed to lower the residual error floor.

## 4 FLATTENED ROTATION tSVD QUANTIZATION

### 4.1 STATISTICAL MOTIVATION OF *Flattened Rotation tSVD Quantization*

Figures 1(a)–(b) plot complementary cumulative distribution functions (CCDFs) for the first-layer weights $\boldsymbol{W}$ and RMSNORM activations $\boldsymbol{X}$ on semi-log axes. Both curves are almost straight, which means that the absolute values decay exponentially and can therefore be modelled by a symmetric Laplace density

$$f(x) = \frac{\lambda}{2}\,e^{-\lambda|x|}.$$

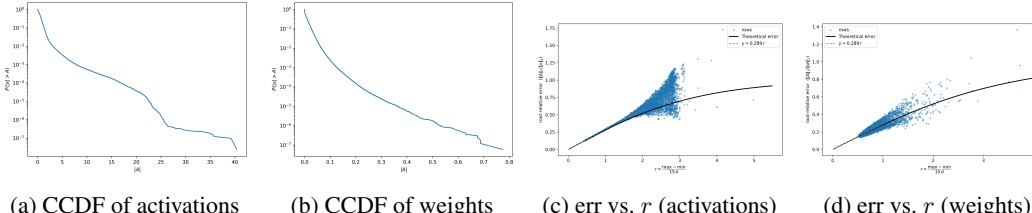

(a) CCDF of activations     (b) CCDF of weights     (c) err vs. $r$ (activations)     (d) err vs. $r$ (weights)

Figure 1: Statistical evidence motivating *FRTQ*. Semi-log complementary CDFs (left pair) reveal near-exponential tails for both activations and weights. Here, *activations* are from LLaMA3-8B, obtained by concatenating all pre-RMSNorm activations (i.e., activations immediately fed into RM-SNorm) across layers; *weights* refer to the layer-0 query projection ($q$) matrix $W_Q$. The right pair shows the row-wise INT4 asymmetric quantization error as a function of $r = (\max - \min)/(15\,\sigma)$ together with the analytic trend $y = r/(2\sqrt{3})$.

For a Laplace random variable $\mathbb{E}[|X|] = 1/\lambda$ and $\sigma = \sqrt{2}/\lambda$, so the standard deviation is a fixed multiple of the average magnitude.

The dynamic range of each row is captured by the single, dimension-free ratio

$$r = \frac{\text{grid length}}{\sigma} = \frac{\max - \min}{15\,\sigma}, \tag{1}$$

where the denominator 15 comes from the $2^4$ INT4 grid $\{-8, \ldots, 7\}$. Assuming the Laplace prior, one can show (see Appendix A.1) that the row-wise INT4 *root* relative error obeys

$$\sqrt{E_{\text{rel}}(r)} = \left[1 - \frac{1}{2}\left((\tfrac{r}{2})^2 + r + 2\right)e^{-r/2} + \frac{1}{2}\,r^3 J_0(r)\,\frac{e^{-r}}{1 - e^{-r}}\right]^{1/2}, \tag{2}$$

with

$$J_0(r) = \int_{-1/2}^{1/2} u^2 e^{-ru}\,du = \frac{(\tfrac{1}{2}r^2 + 4)\sinh(r/2) - 2r\cosh(r/2)}{r^3}. \tag{3}$$

Expanding equation 2 for $r \to 0$ gives $\sqrt{E_{\text{rel}}} \simeq r/(2\sqrt{3}) \approx 0.289\,r$; the error therefore grows linearly with $r$ once the grid is sufficiently fine.

Figures 1(c)–(d) super-impose Eq. equation 2 (solid curves) and its small-$r$ limit (dashed lines) on the empirical INT4 errors for seven first-layer matrices. The theoretical and measured values coincide almost perfectly, confirming that the simple statistic $r$ is a faithful predictor of quantization quality. This parameter can also be applied to measure the different performance between different uniform quantization grids. Because an INT4 step is sixteen times the INT8 step, the linear law immediately implies

$$\sqrt{E_{\text{rel}}^{\text{INT4}}} \approx 16\sqrt{E_{\text{rel}}^{\text{INT8}}},$$

.

Equation equation 1 also explains why a handful of *massive points* can dominate the error: a large $\max - \min$ inflates $r$ and therefore the whole row's quantization loss. Conversely, any operation that shrinks $r$—an orthogonal rotation, a mild rescaling, or the dual-updating strategy introduced in Sections 4.2 and 4.3—immediately lowers the INT4 error. In this sense $r$ provides both a quantitative explanation of why heavy-tailed rows are problematic and a practical handle for suppressing their impact.

### 4.2 ACTIVATION TUNING: UNIFORM PRECONDITIONING BEFORE DFRoT

**Background.** DFRoT Xiang & Zhang (2025) seeks an orthogonal matrix $R$ that minimizes the Frobenius norm $\|X - \mathcal{Q}(XR)\|_2$, where $\mathcal{Q}(\cdot)$ is row-wise INT4 quantization. Each iteration performs an *orthogonal–Procrustes* update against the already quantized target, so every intermediate

matrix is itself a quantization-error–free point. Although this procedure reliably finds a local minimum, the quality of that minimum depends on the landscape, and the landscape is flatter (where error is lower) when the row ratio $r = (\max - \min)/(15\,\sigma)$ is small.

**Uniform Preconditioning.** To push the search into a globally lower region we introduce $K$ *uniform iterations* before the standard DFROT loop (typically $K = 5$). In those preliminary steps the quantized target is *not* $\mathcal{Q}(\boldsymbol{XR})$ but a *uniform–magnitude* matrix

$$\boldsymbol{B} = \mathrm{sign}(\boldsymbol{XR})\,\frac{\|\boldsymbol{XR}\|_{2,\mathrm{row}}}{\sqrt{D}},$$

i.e. every element in a row has the same absolute value while retaining its original sign. This construction collapses the peak-to-std ratio of each row to (almost) unity, thereby driving $r$ towards its theoretical optimum without incurring any quantization error—because $\boldsymbol{B}$ lies exactly on the INT4 grid by design. After $K$ such steps we switch back to the original quantization-aware target and continue with standard DFROT updates until convergence.

**Interpretation.** Rotation lowers quantization noise because it redistributes extreme values; DFROT searches for a low-error orientation by following local gradients. Our Uniform Preconditioning tightens the error bound *before* the search starts, replacing "hill-climbing on rough ground" with "hill-climbing in a valley".

### 4.3 TSVD: OPTIMIZATION ON THE WEIGHT MATRIX

We start from the low-rank split

$$W = L_1 L_2 + R,$$

where $L_2 \in \mathbb{R}^{k \times n}$ spans a $k$-dimensional subspace of the original $n$-dimensional weight space and each row of $L_1 \in \mathbb{R}^{m \times k}$ selects a point in that subspace. Fixing $L_2$, the task for row $i$ is to find a vector $u_i \in \mathbb{R}^k$ that minimises the infinity-norm of the residual

$$r_i = w_i - u_i L_2, \qquad \|r_i\|_\infty = \max_j |r_{ij}|.$$

Intuitively, $L_2$ provides a "palette" of directions; by tuning the coefficients $u_i$ we shift the weight vector inside the subspace so that its projection onto each output dimension never becomes too large. Formally we solve

$$u_i^\star = \arg\min_{u \in \mathbb{R}^k} \|w_i - u L_2\|_\infty,$$

a convex but high-dimensional $\ell_\infty$ regression.

Instead of an expensive exact solution we adopt a greedy iteration. For any row $r$ in the residual matrix $R$, we firstly compute the mean absolute value $m$, and then locate the largest element $r_{j^\star}$, set a target $\tau = 2m\,\mathrm{sign}(r_{j^\star})$ (twice the mean magnitude but with the same sign), and move $u$ so that the updated residual pushes $r_{j^\star}$ towards $\tau$. Two approaches are offered. One is to minimize the 2-norm of the step, and the other is to minimize the 1-norm of the step. We test and accept the step if it strictly reduces $\|r\|_\infty$; otherwise stop iteration. Empirically ten to twenty iterations are enough for most rows, and every accepted update guarantees a drop in the infinity-norm while touching only a single column of $L_2$, hence the procedure remains lightweight.

**Effect on statistics.** The quantization grid $\Delta$ for each row is proportional to $\max_j |r_{ij}|$ after the adjustment, so any reduction in that maximum directly tightens the grid. At the same time the update can *increase* the $\ell_2$-norm of the residual: we trade a few concentrated peaks for a slightly "fatter" body. Assuming the residual entries are approximately symmetric around zero (the usual case after per-row mean subtraction), the standard deviation is

$$\sigma_i = \sqrt{\frac{1}{n}\sum_j r_{ij}^2} = \frac{\|r_i\|_2}{\sqrt{n}}.$$

Hence an increase in $\|r_i\|_2$ implies a commensurate rise in $\sigma_i$. Because $\Delta$ has fallen while $\sigma_i$ has grown, the ratio $\Delta/\sigma_i$—a proxy for relative quantization error—drops significantly. In other words, even though the residual becomes slightly more energetic in $\ell_2$ sense, its most damaging outliers are tamed, yielding smaller uniform-rounding error after quantization.

**Why this complements SVD.** Classical SVD already minimises the Frobenius norm $\|R\|_F$ of the residual, but it is *blind* to individual extremes. Our row-wise tuning therefore attacks the very component—per-token peaks—that standard SVD leaves untouched, providing a simple yet effective second pass that cooperates with the low-rank split to lower quantization error without any additional runtime cost.

## 5 EXPERIMENTS

### 5.1 EXPERIMENT SETTINGS

We build our **FRTQ** implementation upon **DFRot** Xiang & Zhang (2025). To lower computational overhead, we apply dynamic asymmetric per-token quantization to the activations. The KV-cache is quantized asymmetrically with a group size of 128. For the weight matrices, we follow the **GPTQ** framework Frantar et al. (2022), adopting per-channel symmetric quantization and performing a linear search over clipping thresholds to minimize the mean squared error. For calibration, the dataset $\mathcal{X}_{\text{cal}}$ is obtained by sampling a single sequence of length 2048 from the WikiText-2 training set Merity et al. (2016). The rotation matrix $R_1$ is first updated by 5 uniform-preconditioning (UP) steps and then further refined with 100 iterations of DFRot. The resulting $R_1$ is subsequently applied to the model to enforce rotational invariance. In addition, for GPTQ quantization, we prepare a calibration dataset consisting of 128 sequences, each with a length of 2048.

### 5.2 MAIN RESULTS

**Main results.** We evaluate **FRTQ** on perplexity and zero-shot accuracy across the LLaMA/LLaMA2/LLaMA3 family, comparing with **SmoothQuant** Xiao et al. (2022), **GPTQ** Frantar et al. (2022), **OmniQuant** Shao et al. (2023), **SpinQuant** Liu et al. (2025), **OSTQuant** Hu et al. (2025), **QuaRot** Ashkboos et al. (2024), and **DFRot** Xiang & Zhang (2025). Zero-shot performance is computed with the `lm-evaluation-harness` Biderman et al. (2024) (v0.4.5) on nine standard benchmarks—BoolQ Clark et al. (2019), PIQA Bisk et al. (2020), WinoGrande Sakaguchi et al. (2021), OpenBookQA Mihaylov et al. (2018), SocialIQA Sap et al. (2019), HellaSwag Zellers et al. (2019), ARC (Easy/Challenge) Clark et al. (2018), and LAMBADA Paperno et al. (2016); Radford et al. (2019). As summarized in Table 1, **FRTQ** consistently outperforms **DFRot/QuaRot** over models and bit settings: e.g., on **LLaMA3-8B** (W4A4KV16) it reaches **64.23** average accuracy (+0.59 vs. DFRot; +1.51 vs. QuaRot) and lowers perplexity to **9.41** (from 9.58), while on **LLaMA3-70B** it attains **67.45** (+1.17 vs. DFRot) with perplexity **7.25** (from 7.88). Similar gains hold for other models. *Notably, despite approaching the performance of quantization-aware training (QAT) methods such as SpinQuant and OSTQuant, FRTQ is calibration-only: it optimizes a lightweight rotation using a single 2,048-token sequence and applies GPTQ on a small set of 128 sequences—no retraining, no task losses, and no gradients.* This yields substantially lower computational and engineering cost, simpler deployment, and strong, reproducible improvements across 7B–70B scales, offering a practical post-training alternative that recovers much of the QAT benefit at a fraction of the overhead.

### 5.3 ABLATION STUDIES

#### 5.3.1 SUCCESSIVE ABLATIONS

Table 2 tracks accuracy (ZERO-SHOT↑) and perplexity (PPL↓) over three increasingly aggressive stages that start from **DFRot** and culminate in our full pipeline **FRTQ**. The intermediate steps isolate two design choices: *Uniform Preconditioning* (UP) on activations and a subsequent SVD-based split on the weights.

**DFRot → +UP.** Uniform Preconditioning (UP) equalises channel-wise activation scales, directly reducing activation-side rounding error. As expected, part of the error budget is then shifted onto the (still unmodified) weights, so end-to-end quality changes are modest and model-dependent: on Llama3-8B we see a small gain ( +0.39 pp zero-shot; perplexity $-0.14$ ), on L2-7B accuracy ticks up but perplexity is essentially flat to slightly worse ( +0.21 pp; +0.02 ), and on L1-7B both met-

Table 1: Comparison of averaged accuracy on nine Zero-Shot tasks and perplexity on WikiText2. Results for SmoothQuant, GPTQ, OmniQuant, SpinQuant and OSTQuant are from the OSTQuant paper, and DFRot's and QuaRot's results from the official code. * denotes methods that use quantization-aware training to optimize $R_1$.

| #Bits W-A-KV | Method | LLaMA3-8B 0-shot↑ | PPL↓ | LLaMA3-70B 0-shot↑ | PPL↓ | LLaMA2-7B 0-shot↑ | PPL↓ | LLaMA2-13B 0-shot↑ | PPL↓ | LLaMA2-70B 0-shot↑ | PPL↓ | LLaMA-7B 0-shot↑ | PPL↓ | LLaMA-13B 0-shot↑ | PPL↓ | LLaMA-30B 0-shot↑ | PPL↓ |
|---|---|---|---|---|---|---|---|---|---|---|---|---|---|---|---|---|---|
| 16-16-16 | FloatingPoint | 68.09 | 6.14 | 73.81 | 2.86 | 65.21 | 5.47 | 67.61 | 4.88 | 71.59 | 3.32 | 64.48 | 5.68 | 66.67 | 5.09 | 70.00 | 4.10 |
| 4-4-16 | RTN | 33.42 | 6e2 | 31.21 | 8e3 | 32.44 | nan | 30.86 | 8e3 | 30.90 | 7e4 | 32.51 | 7e3 | 31.63 | 3e4 | 31.57 | 2e3 |
| | SmoothQuant | 33.04 | 1e3 | 34.67 | 2e2 | 32.13 | nan | 34.26 | 1e3 | 35.86 | 3e2 | 34.42 | 3e2 | 33.29 | 6e2 | 34.64 | 1e3 |
| | GPTQ | 32.98 | 5e2 | 31.47 | 4e4 | 32.72 | nan | 30.11 | 4e3 | 30.86 | nan | 32.12 | 1e3 | 31.51 | 3e3 | 30.88 | 2e3 |
| | QuaRot | 62.72 | 9.83 | 67.68 | 7.48 | 60.10 | 7.78 | 63.77 | 6.89 | 68.21 | 5.33 | 61.42 | 6.29 | 64.56 | 5.59 | 68.19 | 4.74 |
| | DFRot | 63.64 | 9.58 | 66.28 | 7.88 | 60.49 | 7.74 | 64.01 | 6.71 | 67.55 | 5.30 | 61.69 | 7.70 | 64.50 | 5.59 | 67.96 | 4.80 |
| | **FRTQ** | 64.23 | 9.41 | 67.45 | 7.25 | 61.60 | 7.58 | 64.91 | 6.50 | 68.68 | 5.06 | 61.82 | 6.94 | 65.03 | 5.48 | 67.98 | 4.59 |
| | SpinQuant* | 64.11 | 7.28 | 66.99 | 6.10 | 57.37 | 6.78 | 63.23 | 5.24 | 70.58 | 3.68 | 61.82 | 6.08 | 64.59 | 5.36 | 68.08 | 4.53 |
| | OSTQuant* | 65.14 | 7.24 | 72.21 | 3.97 | 63.90 | 5.60 | 66.24 | 5.14 | 70.92 | 3.57 | 62.72 | 6.04 | 65.80 | 5.40 | 68.52 | 4.43 |
| 4-4-4 | RTN | 33.18 | 7e2 | 30.82 | 8e3 | 32.67 | nan | 30.93 | 7e3 | 31.73 | 7e4 | 32.87 | 1e4 | 31.33 | 3e4 | 31.64 | 2e3 |
| | SmoothQuant | 32.96 | 1e3 | 33.76 | 3e2 | 32.12 | nan | 33.36 | 1e3 | 35.54 | 3e2 | 33.32 | 3e2 | 33.28 | 5e2 | 34.65 | 1e3 |
| | GPTQ | 33.71 | 6e2 | 31.20 | 4e4 | 33.52 | nan | 27.85 | 5e3 | 31.09 | nan | 31.80 | 2e3 | 30.63 | 3e3 | 31.07 | 2e3 |
| | OmniQuant | 32.33 | 4e2 | – | – | 48.40 | 14.26 | 50.35 | 12.30 | – | – | 48.46 | 11.26 | 45.63 | 10.87 | 45.04 | 12.35 |
| | QuaRot | 62.88 | 9.83 | 67.89 | 7.49 | 60.46 | 7.83 | 63.56 | 6.90 | 68.11 | 5.34 | 61.34 | 6.31 | 64.85 | 5.59 | 67.90 | 4.74 |
| | DFRot | 63.50 | 9.60 | 65.99 | 7.84 | 60.70 | 7.87 | 64.30 | 6.73 | 67.54 | 5.30 | 61.50 | 6.29 | 64.29 | 5.60 | 67.54 | 4.81 |
| | **FRTQ** | 63.78 | 9.40 | 67.28 | 7.24 | 61.36 | 7.58 | 65.36 | 6.53 | 68.79 | 5.07 | 61.74 | 6.19 | 65.11 | 5.48 | 68.08 | 4.60 |
| | SpinQuant* | 64.10 | 7.35 | 66.31 | 6.24 | 62.01 | 5.96 | 64.13 | 5.74 | 70.57 | 3.61 | 61.32 | 6.12 | 64.95 | 5.39 | 68.14 | 4.55 |
| | OSTQuant* | 65.37 | 7.29 | 71.69 | 4.01 | 63.18 | 5.91 | 65.41 | 5.25 | 70.84 | 3.59 | 62.55 | 6.07 | 65.43 | 5.40 | 68.20 | 4.42 |

Table 2: Ablation studies of FRTQ.

| Method | Llama3-8B Zero-shot(↑) | ppl(↓) | Llama2-7B Zero-shot(↑) | ppl(↓) | Llama1-7B Zero-shot(↑) | ppl(↓) |
|---|---|---|---|---|---|---|
| DFRot | 63.64 | 9.582 | 60.49 | 7.735 | 61.69 | 7.073 |
| +UP | 64.03 | 9.439 | 60.70 | 7.752 | 61.31 | 7.108 |
| +UP+SVD | 63.54 | 9.402 | 60.75 | 7.626 | 61.74 | 6.928 |
| FRTQ | 64.23 | 9.410 | 61.60 | 7.577 | 61.82 | 6.936 |

rics move slightly against us ($-0.38$ pp; $+0.04$). This trade-off is intentional: UP first cleans up activation error while deferring the weight-side correction to the subsequent SVD stage.

**+UP $\rightarrow$ +UP+SVD.** Adding a rank-$k$ SVD branch pulls dominant weight directions into full precision, tightening the residual grid and making better use of UP's lower variance. Perplexity improves on all three models (about $-0.04$ to $-0.18$), while accuracy is mixed: it rises on L1-7B ($+0.43$ pp) and is essentially flat on L2-7B ($+0.05$ pp), but dips on Llama3-8B ($-0.49$ pp). Overall, this step delivers consistent perplexity gains and stabilises the UP hand-off.

**+UP+SVD $\rightarrow$ FRTQ (tSVD).** FRTQ replaces the vanilla SVD with the tuned variant from Section 4.3, which further depresses per-row maxima and lowers the GSR via a row-wise $\ell_\infty$ objective. Despite requiring only a short calibration, tSVD lifts zero-shot accuracy across all models ($+0.69$ pp on Llama3-8B, $+0.85$ pp on L2-7B, $+0.08$ pp on L1-7B), while perplexity is essentially unchanged on Llama3-8B/L1-7B ($\pm0.01$) and modestly better on L2-7B ($-0.05$). In practice, this final tuning recovers most of the remaining accuracy headroom with minimal extra compute.

### 5.3.2 EXPERIMENTS ON UNIFORM PRECONDITIONING

To gauge the specific impact of *Uniform Preconditioning* (UP) we track two statistics across seven LLaMA-family checkpoints. The **grid–std ratio** (GSR) is computed as the quantization grid size $\Delta$ divided by the residual's standard deviation $\sigma$; the **quantization error** (QErr) is the mean-squared rounding error under W4A4 activation quantization. Both metrics are first computed *row-wise* and then averaged across all rows (and layers) to obtain the values reported below.

At initialization, GSR values can reach two orders of magnitude (e.g. 101.9 on LLaMA2-70B), implying coarse grids and large rounding errors. Applying a random orthogonal rotation already cuts GSR by roughly $5\times$, and a structured Hadamard rotation delivers a similar benefit with slightly smaller error. Data-driven DFRot trims another five to seven percent from QErr, showing that even small, task-aware adjustments help.

Uniform Preconditioning produces a dramatic change: across all models, GSR falls by a further $20\%$–$25\%$, and QErr drops into the $\sim0.30$ range. The near-linear correspondence between GSR

Table 3: GSR and QErr at successive stages of activation conditioning.

| Model | Init | | +Orth | | +Hadamard | | +DFRot | | +UP | |
|---|---|---|---|---|---|---|---|---|---|---|
| | GSR | QErr | GSR | QErr | GSR | QErr | GSR | QErr | GSR | QErr |
| LLaMA3-8B | 22.36 | 0.650 | 4.74 | 0.373 | 4.70 | 0.372 | 4.77 | 0.349 | **3.28** | **0.322** |
| LLaMA2-7B | 40.42 | 0.704 | 4.79 | 0.375 | 4.56 | 0.363 | 4.58 | 0.360 | **3.68** | **0.308** |
| LLaMA-7B | 42.52 | 0.726 | 4.74 | 0.373 | 4.47 | 0.359 | 4.38 | 0.358 | **3.65** | **0.302** |
| LLaMA-13B | 55.36 | 0.715 | 4.83 | 0.377 | 4.42 | 0.358 | 4.45 | 0.359 | **3.72** | **0.305** |
| LLaMA-30B | 57.42 | 0.712 | 4.92 | 0.381 | 4.60 | 0.364 | 4.49 | 0.358 | **3.75** | **0.305** |
| LLaMA2-13B | 33.09 | 0.726 | 4.79 | 0.397 | 4.74 | 0.395 | 4.73 | 0.394 | **3.74** | **0.307** |
| LLaMA2-70B | 101.88 | 0.776 | 4.94 | 0.404 | 4.33 | 0.383 | 4.35 | 0.357 | **3.10** | **0.313** |
| LLaMA3-70B | 34.71 | 0.727 | 4.95 | 0.404 | 4.93 | 0.404 | 4.99 | 0.380 | **3.10** | **0.316** |

and QErr across stages confirms that tightening the grid relative to the residual's spread directly lowers quantization error. Although UP alone does not guarantee higher end-to-end accuracy ( Section 5.3.1), it establishes a much lower activation error floor, setting the stage for the weight-side refinements explored in the next sections.

Finally, GSR also makes a well-known empirical point explicit: the structured Hadamard rotation tends to outperform a random orthogonal rotation, as evidenced by consistently lower post-Hadamard GSR.

## 5.4 EXPERIMENTS ON TSVD

We tracked *all* linear weight matrices for three models and report per-model averages under **Plain** (original weights), **SVD**, and **tSVD**). The quantization error is computed on the residual and does not account for the fact that the SVD moves part of the energy into floating-point factors.

Table 4: GSR, INT4 quantization error (QErr), and their ratio for the original weight matrix and the residuals after SVD and tSVD.

| | GSR | | | INT4 error | | | GSR / error | | |
|---|---|---|---|---|---|---|---|---|---|
| Model | Plain | SVD | tSVD | Plain | SVD | tSVD | Plain | SVD | tSVD |
| Llama3-8B | 0.531 | 0.530 | 0.497 | 0.161 | 0.161 | 0.145 | 3.290 | 3.294 | 3.434 |
| Llama2-7B | 0.528 | 0.528 | 0.494 | 0.160 | 0.160 | 0.144 | 3.293 | 3.292 | 3.433 |
| Llama-7B | 0.528 | 0.528 | 0.473 | 0.160 | 0.160 | 0.137 | 3.293 | 3.294 | 3.446 |

SVD alone essentially mirrors **Plain**. With the tuned scalings we deliberately push down the per-row maxima; since the quantization grid width is set by this maximum, the effective grid tightens, *GSR* decreases, and the INT4 error drops in lockstep across checkpoints. The normalized ratio GSR/error stays close to the high-resolution constant $2\sqrt{3} \approx 3.464$: it starts slightly below this under Plain/SVD and increases after tSVD because the maxima are explicitly adjusted. Crucially, even as this ratio increases, the final quantization error still decreases markedly.

## 5.5 DO ROTATIONS AND TUNINGS AFFECT THE ELEMENT-WISE DISTRIBUTION?

Rotating the weights—whether by a Hadamard matrix, a random orthogonal transform, or a tuned rotation—*reshapes* their one–dimensional marginal. Empirically, a data–free Hadamard step turns the original double–exponential (Laplace–like) heavy tails into a noticeably *shorter–tailed* law, as seen from the CCDF in Fig. 2a.

Not all rotations are equal in their quantization footprint. Although both DFRot and FRTQ are rotation-based, FRTQ's tuned branch further suppresses per–row maxima while preserving row-wise RMS, thereby shrinking the per–row grid–to–scale ratio $r = \Delta/\sigma$. Consequently, in Fig. 2b the FRTQ cloud sits systematically *below and to the left* of DFRot—its GSR is smaller and its quantization error is lower.

Why does the high–resolution law still hold even after the distribution changes? Our measurements show that rotations do alter the marginal law—heavy tails become much shorter—yet the leading

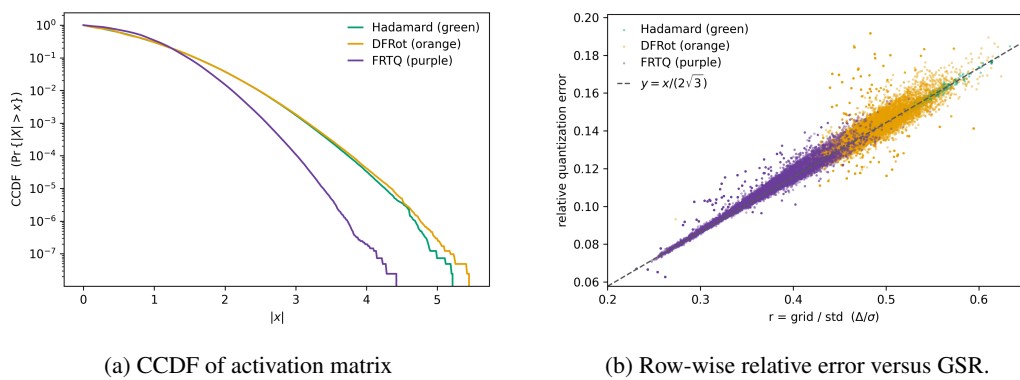

(a) CCDF of activation matrix

(b) Row-wise relative error versus GSR.

Figure 2: Element-wise distributions and quantization errors of the activation matrix under three schemes—Hadamard transform, DFRot, and *FRTQ*. The activations are rotated version of the initial activation matrix in Figure 1.

coefficient of the high–resolution expansion remains the same. Motivated by this, we revisited the classical analysis and verified that, for any absolutely continuous source with finite variance and mild smoothness, the normalized RMS error admits

$$\sqrt{E_{\mathrm{rel}}(r)} \;=\; \frac{r}{2\sqrt{3}} \;+\; O(r^3), \qquad r = \Delta/\sigma, \;\; r \to 0,$$

and that the coefficient $1/(2\sqrt{3})$ is *distribution–agnostic*: it depends only on cell geometry (the second moment of a uniform variable over a quantization interval), not on the detailed shape of the input density. Orthogonal transformations leave $\sigma = \sqrt{\mathbb{E}[X^2]}$ unchanged, so they preserve the first–order slope; what changes across rotations is the operating point $r = \Delta/\sigma$ through the way mass is redistributed (and, under tuning, through a reduction of the effective $\Delta$). A more detailed derivation and discussion are provided in Appendix A.2.

This perspective explains the empirical plots: DFRot and FRTQ both align with the universal $1/(2\sqrt{3})$ line in the fine–grid regime, but FRTQ lands *at a smaller $r$* (lower GSR) and therefore achieves *lower error*. Practically, this means we can tune quite aggressively: whenever the per–row maximum is pushed down, the grid tightens, and across a wide range of distributional shapes the quantization error drops accordingly.

## 6 Conclusion

Assuming activations and weights follow double–exponential laws, the relative quantization error scales linearly with the grid–std ratio (GSR) when GSR is small, i.e., $\propto \Delta/\sigma$. Thus: *shrink the grid, enlarge the standard deviation.* We exploit two PTQ degrees of freedom—orthogonal rotations for activations and low-rank decompositions for weights. Uniform Preconditioning (UP) iteratively learns a rotation that minimizes row-wise GSR; tuned SVD (tSVD) adjusts the low-rank branch to suppress peak residuals per weight row.

This pipeline retains PTQ strengths while avoiding QAT's heavy optimization: all updates are closed-form or one-pass, with no back-propagation, labels, or lengthy calibration. Across LLaMA models, UP+tSVD yields sizable perplexity and zero-shot gains, approaching QAT quality at a fraction of compute and latency by systematically reducing GSR for activations and weights.

We report perplexity and zero-shot accuracy, but these can miss behavioral drift. We advocate *distance-to-baseline* diagnostics—e.g., KL divergence on held-out token distributions and prediction-flip rates on multiple-choice tasks Dutta et al. (2024)—as complementary, and view comprehensive D2B evaluation as valuable future work.

Empirically, a data-free Hadamard rotation shortens the Laplace-like tails of weight distributions; iterating it further lightens tails, with two passes outperforming one. Repeated random Hadamards appear to drive elementwise marginals toward an approximately Gaussian regime, which is quantization-friendly. These observations are preliminary and merit systematic study.

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

# A APPENDIX

## A.1 QUANTIZATION ERROR FOR LAPLACE–DISTRIBUTION NORMALISED BY THE STANDARD DEVIATION

**1. Prior and scale parameters.** Assume every element of the row vector $x \in \mathbb{R}^n$ follows a zero–mean Laplace distribution

$$f(x) = \frac{\lambda}{2} e^{-\lambda |x|}, \qquad x \in \mathbb{R}, \ \lambda > 0.$$

Its key moments are

$$\mathbb{E}[X] = 0, \qquad \mathbb{E}[|X|] = \frac{1}{\lambda}, \qquad \sigma = \sqrt{\text{Var}[X]} = \frac{\sqrt{2}}{\lambda}.$$

**2. Peak–to–$\sigma$ ratio and grid length.** Define the deterministic peak

$$R := \frac{\max |x_i|}{\sigma} \ (> 0), \qquad M = R\,\sigma.$$

For symmetric INT4 uniform quantization $(-8, \ldots, 7)$ the grid length is

$$d = \frac{M}{7} = \frac{R}{7}\,\sigma, \qquad r := \frac{d}{\sigma} = \frac{R}{7},$$

so that $\lambda d = \sqrt{2}\,r$.

**3. Decision boundaries and representatives.** Quantizer centres and boundaries are

$$q_k = k\,d, \qquad b_k = \left(k - \tfrac{1}{2}\right)d, \qquad k = 0, 1, 2, \ldots, \ b_0 = 0.$$

**4. Mean–square error per cell.** Let $I_k = \int_{b_k}^{b_{k+1}} (x - q_k)^2 f(x)\,dx$.

*(a) First cell, $k = 0$.* With $A = d/2$ and $\lambda A = r/2$,

$$I_0 = \frac{\sigma^2}{2} \left[ 2 - \left( (\tfrac{r}{2})^2 + r + 2 \right) e^{-r/2} \right].$$

*(b) Generic cell, $k \geq 1$.* Substitute $x = kd + ud \ (u \in [-\tfrac{1}{2}, \tfrac{1}{2}])$,

$$I_k = \frac{\sigma^2}{2}\, r^2\, e^{-rk}\, J_0(r), \qquad J_0(r) = \int_{-1/2}^{1/2} u^2 e^{-ru}\,du = \frac{(\tfrac{1}{2} r^2 + 4)\sinh(r/2) - 2r\cosh(r/2)}{r^3}.$$

**5. Total MSE.** Because $I_k \propto e^{-rk}$ for $k \geq 1$,

$$\sum_{k=1}^{\infty} I_k = \frac{\sigma^2}{2}\, r^2 J_0(r)\, \frac{e^{-r}}{1 - e^{-r}}.$$

Hence

$$E_{\text{err}} = I_0 + \sum_{k \geq 1} I_k = \frac{\sigma^2}{2} \Big[ 2 - \left( (\tfrac{r}{2})^2 + r + 2 \right) e^{-r/2} + r^2 J_0(r)\, \frac{e^{-r}}{1 - e^{-r}} \Big].$$

**6. Normalised error and small–grid limit.** Normalising by $\mathbb{E}[X^2] = \sigma^2$ and taking the square root yields

$$\sqrt{E_{\mathrm{rel}}(r)} = \sqrt{1 - \frac{1}{2}\big((\tfrac{r}{2})^2 + r + 2\big)e^{-r/2} + \frac{1}{2}\, r^3 J_0(r)\, \frac{e^{-r}}{1 - e^{-r}}}$$

Expanding for $r \to 0$,

$$E_{\mathrm{rel}}(r) = \frac{r^2}{12} - \frac{7r^4}{2880} + O(r^5), \qquad \sqrt{E_{\mathrm{rel}}(r)} = \frac{r}{2\sqrt{3}} + O(r^3) \approx 0.289\, r.$$

Thus, when the grid is much finer than the standard deviation, the relative $\ell_2$ quantization error increases linearly with slope $1/(2\sqrt{3})$.

A.2  UNIVERSALITY OF THE HIGH–RESOLUTION SLOPE FOR SMOOTH SOURCES

**1. Setup.** Let $X$ be a zero–mean, continuous random variable with finite variance $\sigma^2 = \mathrm{Var}[X]$ and pdf $f(x)$. For an infinite, symmetric, uniform quantizer with step $\Delta = r\sigma$ $(r > 0)$ we set

$$q_k = k\Delta, \qquad b_k = \big(k - \tfrac{1}{2}\big)\Delta, \qquad k \in \mathbb{Z}, \qquad Q(X) = q_k \ \text{if} \ b_k \le X < b_{k+1}.$$

**2. Decomposition of the MSE.** Writing the cell error $e(x) = x - Q(x)$ $(|e| \le \Delta/2)$,

$$E_{\mathrm{err}} = \sum_{k=-\infty}^{\infty} \int_{b_k}^{b_{k+1}} e(x)^2\, f(x)\, dx.$$

**3. High–resolution approximation $(r \ll 1)$.** When the grid is much narrower than the scale over which $f$ changes, $f(x) \approx f(k\Delta)$ inside the $k$-th cell; hence

$$\int_{b_k}^{b_{k+1}} e(x)^2 f(x)\, dx \ \approx \ f(k\Delta) \int_{-\Delta/2}^{\Delta/2} e^2\, de \ = \ f(k\Delta)\, \frac{\Delta^3}{12}.$$

The Riemann sum $\sum_k f(k\Delta)\Delta$ converges to $\int_{-\infty}^{\infty} f(x)\, dx = 1$, so

$$E_{\mathrm{err}} \ \approx \ \frac{\Delta^2}{12}.$$

**4. Relative error and the universal slope.** By our assumption, the distribution is symmetric and thus the energy of the distribution is exactly the variation $\sigma^2$. Dividing by $\sigma^2$ gives

$$E_{\mathrm{rel}}(r) = \frac{E_{\mathrm{err}}}{\sigma^2} \ \approx \ \frac{\Delta^2/12}{\sigma^2} = \frac{r^2}{12}, \qquad \sqrt{E_{\mathrm{rel}}(r)} \ \approx \ \frac{r}{2\sqrt{3}}.$$

The factor $1/(2\sqrt{3})$ is precisely $\sqrt{\mathbb{E}[U^2]}$ with $U \sim \mathrm{Unif}\big(-\tfrac{1}{2}, \tfrac{1}{2}\big)$; it reflects the geometry of uniform quantization and is *independent* of the source pdf.

## A.3 PSEUDO CODE FOR FRTQ

---

**Algorithm 1:** FRTQ-A — UP $\rightarrow$ DFRot for activations (INT4)

**Input:** Calibration matrix $\mathcal{X} \in \mathbb{R}^{N \times d}$; bit-width $b$=4; UP passes $K$=5; DFRot passes $T$=100

**Output:** Rotation $R^\star \in \mathrm{O}(d)$

1   $R \leftarrow \text{HADAMARDINIT}(d)$;

2   **Phase 1: Uniform-Preconditioning**

3   **for** $k \leftarrow 1$ **to** $K$ **do**

4      $X \leftarrow \mathcal{X}R$;

5      $c \leftarrow \|X\|_{2,\text{row}}/\sqrt{d}$;

6      $B \leftarrow \text{sign}(X) \odot c$;

7      $(U, \_, V) \leftarrow \text{SVD}(X^\top B)$;;

8      $R \leftarrow R(UV^\top)$;

9   **end**

10   **Phase 2: DFRot Refinement**

11   **for** $t \leftarrow 1$ **to** $T$ **do**

12      $X \leftarrow \mathcal{X}R$;

13      $B \leftarrow Q_b(X)$;              // INT4 asymmetric quantization

14      $(U, \_, V) \leftarrow \text{SVD}(X^\top B)$;;

15      $R \leftarrow R(UV^\top)$;

16   **end**

17   **return** $R^\star \leftarrow R$

---

**Algorithm 2:** FRTQ-W — row-wise tSVD refinement

**Input:** $W \in \mathbb{R}^{m \times n}$, rank $r$, inner iters $k_{\max}$, overshoot $k$, step $\alpha$

**Output:** $L_1 \in \mathbb{R}^{m \times r}$, $L_2 \in \mathbb{R}^{r \times n}$

1   $(U, \Sigma, V^\top) \leftarrow \text{SVD}(W)$; $L_1 \leftarrow U\Sigma$; $L_2 \leftarrow V^\top$; $R \leftarrow W - L_1 L_2$

2   **for** $i = 1$ **to** $m$ **do**

3      $r \leftarrow R[i,:]$

4      **for** $t = 1$ **to** $k_{\max}$ **do**

5          $\mu \leftarrow \text{mean}(|r|)$; $j \leftarrow \arg\max |r|$; $r_{\max} \leftarrow r_j$

6          **if** $|r_{\max}| \leq k\mu$ **then**

7              **break**

8          **end**

9          $\tau \leftarrow 2\mu \, \text{sign}(r_{\max})$; $v \leftarrow L_2[:,j]$; $\delta \leftarrow -\dfrac{r_{\max} - \tau}{v^\top v} \, v$; $r' \leftarrow r - \alpha \, \delta L_2$

10         **if** $\|r'\|_\infty < \|r\|_\infty$ **then**

11            $L_1[i,:] \leftarrow L_1[i,:] + \alpha\delta$; $r \leftarrow r'$; **continue**

12         **end**

13         $c \leftarrow \arg\max |v|$; $\delta \leftarrow \mathbf{0}_r$; $\delta_c \leftarrow -\dfrac{r_{\max} - \tau}{v_c}$; $r' \leftarrow r - \alpha \, \delta L_2$

14         **if** $\|r'\|_\infty < \|r\|_\infty$ **then**

15           $L_1[i,:] \leftarrow L_1[i,:] + \alpha\delta$; $r \leftarrow r'$

16         **end**

17         **else**

18           **break**

19         **end**

20      **end**

21      $R[i,:] \leftarrow r$

22   **end**

23   **return** $L_1, L_2, R$

---

## APPENDIX D    USE OF LARGE-LANGUAGE MODELS

**Disclosure.** We employed an LLM only in two limited ways:

- **Language polishing:** improving grammar, wording, and LaTeX phrasing;
- **Minor scripting:** generating small one-off data–conversion scripts, each manually verified.

All core ideas, algorithmic designs (UP, tSVD), and experimental protocols were conceived, implemented, and checked by the authors. No confidential material was provided to an external model, and no hidden prompt injections or similar manipulations were used.

**Policy compliance.** Our usage conforms to the two ICLR 2026 LLM rules:

1. *Disclosure.* Every LLM contribution is acknowledged here and in the submission form.
2. *Responsibility.* The authors have manually inspected all LLM-generated text or code and remain fully accountable for the final content. Hence any potential hallucination, factual error, or ethical breach has been screened out by human review.

