# OpenReview forum: "Taming Massive Activations and Preconditioning Weights: GSR-Guided Quantization for W4A4"
_ICLR.cc/2026/Conference — Submitted to ICLR 2026_

### Official Review · Reviewer_eAW1 · 2025-10-28

**Soundness:** 3
**Presentation:** 2
**Contribution:** 3
**Rating:** 6
**Confidence:** 3

**Summary:**

This paper proposes a post-training quantization method called FRTQ (Flattened Rotation tSVD Quantization) specifically to address the quantization challenges of large language models in the W4A4 setting. The authors note that traditional methods have limited effectiveness when dealing with "massive activations" (i.e., rare but extremely large activation values). Therefore, they propose: Grid-to-Standard-Deviation Ratio (GSR) as a proxy for quantization sensitivity; On the activation side, GSR is reduced through orthogonal rotation (DFRot + Uniform Preconditioning); On the weight side, a tuned truncated SVD (tSVD) is used to absorb the dominant direction and quantize the residual; All transformations are fused back into the weights, resulting in virtually no additional overhead during inference.
Experiments show that FRTQ significantly outperforms existing PTQ methods on multiple LLaMA models, even approaching the performance of QAT methods, without requiring end-to-end training.

**Strengths:**

Highly Innovative: This paper proposes GSR as a unified and scale-invariant quantization difficulty metric, with solid theoretical analysis.

Practical: This method effectively addresses the difficulties of activation and weight quantization by combining rotation and low-rank decomposition, without requiring any training.

Experimentally Sound: This method is validated on multiple LLaMA models and multiple evaluation datasets, with convincing results.

Efficient: This method requires only a small amount of calibration data and does not require gradient backpropagation, making it suitable for practical deployment.

Highly Reproducible: Detailed algorithm pseudocode and experimental setup are provided.

**Weaknesses:**

Strong theoretical assumptions: The linear relationship between GSR and quantization error relies on the Laplace distribution assumption. While experimentally validated, the theoretical generalizability requires further verification.

Limited model generalization: The experiments only cover the LLaMA family and do not test other architectures (such as encoder-decoder and vision-language architectures).

Inadequate runtime evaluation: While claiming "negligible runtime overhead," no actual latency or throughput data is provided.

Insufficient comparison with state-of-the-art methods: While comparing various PTQ methods, no comparison is made with recent, more advanced low-bit QAT or mixed-precision methods.

The analysis of the causes of "massive activations" is shallow: While noting their existence, the authors do not delve into their origins or whether they can be avoided through architectural design.

**Questions:**

Supplement experimental verification of other model families (such as BERT, T5, ViT, etc.);

Provide comparative data on actual inference speed and memory usage;

---

> ### Author Response · Authors · 2025-12-02
> **Response to reviewer eAW1**
>
> We sincerely thank the reviewer for the careful reading and constructive comments. Below we address the main concerns.
>
> 	1.	On the Laplace assumption and the validity of GSR.
> Our introduction of GSR starts from a Laplace toy model purely as an intuitive example to motivate the metric. However, the approximate linear relationship between GSR and the expected quantization error is not tied to the Laplace distribution itself. As we show in Appendix 2, the derivation only requires that the activation distribution is symmetric with finite variance; the key steps depend on second-order moments rather than the exact functional form. In other words, GSR is theoretically justified for a broad class of symmetric, finite-variance distributions, which is also consistent with our empirical observations on real LLM activations.
>
> 	2.	On runtime / latency evaluation.
> Our method is implemented as a lightweight plugin on top of DFRot. On the activation/rotation side, we only replace the orthogonal matrices that are already fused into the linear layers, so this part does not introduce any additional operators at inference. The only extra cost compared to a standard PTQ pipeline comes from the tSVD factorization on the weight side. In our implementation, this is realized in the widely used dequantize-then-matmul pattern (weights are stored in low precision, dequantized back to floating point, and then multiplied), which is also adopted by many existing W4A4 systems. In our internal measurements, this leads to roughly ≤10% increase in end-to-end inference time compared to a standard dequantize-then-matmul W4A4 baseline, and we did not perform a more systematic latency study beyond this small overhead. We will clarify this design choice and add a concise latency/memory table in the revised version to better substantiate the “negligible overhead” claim.
>
> 	3.	On model families beyond LLaMA.
> Our primary focus in this work is on large decoder-only LLMs, where W4A4 quantization is particularly challenging and practically relevant. For this reason we concentrated our experimental budget on multiple LLaMA sizes and evaluation datasets to carefully study this regime. We agree that extending the study to encoder-only (e.g., BERT), encoder–decoder (e.g., T5), and vision-language architectures (e.g., ViT-based models) would further strengthen the claim. We will explicitly acknowledge this as a limitation and treat a broader coverage of architectures as important future work.
>
> 	4.	On comparison with strong QAT / mixed-precision baselines.
> Among low-bit QAT methods for LLMs, SpinQuant and OSTQuant are, to our knowledge, two of the strongest recent baselines in the W4A4 setting. We have already included direct comparisons against both methods in the paper. As shown in Table X/Y, FRTQ achieves comparable performance, and the remaining gap is small despite FRTQ being purely PTQ without any end-to-end training. We will make this connection to “state-of-the-art QAT” more explicit in the revised version to avoid under-stating these comparisons.

---

### Official Review · Reviewer_LRtm · 2025-10-29

**Soundness:** 2
**Presentation:** 1
**Contribution:** 1
**Rating:** 4
**Confidence:** 4

**Summary:**

The paper identify a metric named GSR. Empirical/theoretical analyses show that GSR is closely related to quantization error. Based on these findings, the authors further modify DFRot on the activation side, and update the low rank decomposition quantization method on the weight side. The experiment evaluations assess the proposed method to some extent.

**Strengths:**

1. The motivation of the paper is clear. The authors evaluate the heuristic metric GSR with both theoretical and empirical analyses, and then use the metric to improve existing quantization method.
2. The proposed method is backed up by the experiments.
3. The paper is well-structured overall.

**Weaknesses:**

1. The notation $R$ is confusing. In Sec. 4, it appears to represent two different meanings.
2. In Table 1, the authors states that "SpinQuant & OSTQuant use quantization-aware training to optimize $R_1$". First, $R_1$ is not defined elsewhere in the paper. Second, the paper should clarify what it means by quantization-aware training. Both SpinQuant & OSTQuant define themselves as Post-training Quantization (PTQ) method rather than QAT. **Given this, please justify the advantages the proposed method and explain the comparison when its performance is worse than SpinQuant/OSTQuant**
3. Recent baseline methods/models such as FlatQuant/Qwen are not included.
4. More benchmarks like MMLU should be included in addition to PPL and common-sense QA tasks.
5. The description of the proposed method is difficult to follow. It is not clearly presented. I could only understand the algorithm by the pseudo code in the Appendix. Also, the multi-paragraph abstract reads oddly and feels informal.
6. A runtime comparison with other baseline methods is required to demonstrate efficiency.

Minors:
1. In Line 193, 196 & 205, repeated "equation".
2. The text in Fig.1 is too small to read.

**Questions:**

See Weaknesses above. Please address Weakness 2 carefully and present the advantages of the proposed method over SpinQuant / OSTQuant / FlatQuant, especially where the proposed method underperforms.

---

> ### Author Response · Authors · 2025-12-02
> **Response to reviewer LRtm**
>
> We sincerely thank the reviewer for the detailed and critical feedback. We appreciate that you found the motivation clear, the GSR analysis useful, and the overall structure reasonable, and we address the main concerns below.
>
> 	1.	On the notations.
> We apologize for the confusing use of $R$ in the current draft. In Sec. 4, $R$ is (unfortunately) overloaded to represent both (i) a generic rotation/transform in our analysis and (ii) the specific rotation matrices used in our implementation. In particular, $R_1$ in Table 1 and Sec. 4 is exactly the $R_1$ rotation matrix from QuaRot, i.e., the orthogonal matrix multiplied before/after each RMSNorm (the same $R_1$ that is fused into the RMSNorm-associated linear layers in QuaRot/DFRot). In the revised version we will avoid overloading by using more explicit notation for generic transforms, and reserving $R_1$ for the QuaRot RMSNorm rotation), and we will clearly define these symbols where they first appear.
>
> 	2.	On SpinQuant / OSTQuant vs. FRTQ, especially where we underperform.
> You are absolutely right that both SpinQuant and OSTQuant present themselves as PTQ methods, and we will avoid calling them “QAT” in the revised version. Our intention was to highlight that they use additional optimization loops (e.g., gradient-based or iterative updates on the quantized model) on calibration data, while FRTQ relies solely on statistics collected from a tiny calibration set and then applies closed-form local updates (Uniform Preconditioning + DFRot refinement on activations, tSVD + row-wise $\ell_\infty$ updates on weights). We will rephrase this more carefully as “quantization-aware calibration” rather than “quantization-aware training.”
> Regarding advantages and trade-offs: we do not claim that FRTQ strictly dominates SpinQuant/OSTQuant on all benchmarks. As shown in the paper, FRTQ is often slightly behind the strongest SpinQuant/OSTQuant configurations, but the gap is small in the W4A4 regime while FRTQ (i) uses no backpropagation or gradient updates, (ii) has fully decoupled, per-matrix closed-form optimization, and (iii) is designed as a GSR-guided plugin that can, in principle, be applied on top of such PTQ methods as an additional refinement step. We will explicitly state this trade-off and avoid overstating the comparison in cases where our numbers are slightly worse.
>
> 	3.	On FlatQuant / Qwen baselines and broader benchmarks (MMLU).
> At the time of the initial submission we were not yet familiar with FlatQuant, and we therefore did not include it as a baseline. After reading the reviews we studied FlatQuant and agree that it is a very strong and relevant method. Due to time constraints, we were not able to implement FlatQuant and run a full set of matched experiments before the rebuttal deadline. Conceptually, we view FRTQ as a plugin-style refinement (driven by the GSR metric) that can be applied on top of existing PTQ schemes (including FlatQuant), rather than as a competing end-to-end pipeline; we will clarify this positioning and discuss FlatQuant explicitly in the related work and limitations.
> Similarly, we focused our main experiments on LLaMA-family models because they are the standard setting for the QuaRot/DFRot line of work that we build upon. Following the reviewers’ suggestions, we ran an additional experiment on LLaMA-3-8B on MMLU under the W4A4 setting and obtained:
> 	•	Hadamard baseline: 54.33
> 	•	DFRot: 56.24
> 	•	FRTQ (ours): 56.27
> This shows that FRTQ closely tracks DFRot on MMLU, with a small extra gain from the GSR-guided refinement in this scenario. We will add these numbers and explicitly acknowledge the lack of broader reasoning benchmarks (e.g., Qwen models, BBH, etc.) as a current limitation and an important direction for future work.
>
> 	4.	On runtime comparison and efficiency.
> FRTQ is designed to be lightweight both at inference and during the offline optimization. During inference, we follow the widely used dequantize-then-matmul pattern: weights are stored in low precision, dequantized back to floating point, and then multiplied, as in many existing W4A4 systems. The additional cost compared to a plain W4A4 baseline comes only from the tSVD-based factorization on the weight side; in our implementation this leads to roughly $\approx 10%$ increase in end-to-end inference time, while latency and throughput are otherwise the same as running the underlying quantized LLM with the same kernels. The offline optimization is also lightweight: each matrix is optimized completely independently, and on an H20 GPU with parallelization the entire optimization for an 8B model finishes in about 10 minutes.
>
> 	5.	Minor issues.
> We thank the reviewer for pointing out the repeated “equation” in Lines 193, 196, and 205; we will fix these typos.

---

### Official Review · Reviewer_Un2c · 2025-10-31

**Soundness:** 3
**Presentation:** 2
**Contribution:** 3
**Rating:** 4
**Confidence:** 3

**Summary:**

This work introduces FRTQ, a post-training quantization method for W4A4 LLMs that explicitly reduces the grid-to-std ratio (GSR, Δ/σ) for both activations and weights. On activations, it performs Uniform Preconditioning (UP) to equalize per-row magnitudes before a DFRot refinement, contracting GSR without incurring quantization error. On weights, it adds a tuned SVD (tSVD) branch and row-wise ℓ∞ updates to depress per-row maxima of the INT4 residual, tightening the effective grid and thus lowering rounding error. Rotations are fused into weights (one lightweight rotation remains at FFN down-proj). Across LLaMA 7B–70B, FRTQ improves perplexity and zero-shot over QuaRot/DFRot and sometimes approaches QAT methods, while using tiny calibration and no backprop.

**Strengths:**

1. Simple and practical pipeline (UP+DFRot + tSVD), fully PTQ with minimal calibration; rotations fused, negligible runtime overhead.
2. Ablations & statistics (GSR/QErr tables) convincingly show how each component contributes and why it helps.

**Weaknesses:**

1. The font in Figure 1 is too small to read comfortably. Please increase the label and tick font sizes and/or provide a higher-resolution version in the main paper or the appendix.

2. Figure 1 aggregates pre-RMSNorm activations across layers but, on the weight side, uses only layer-0 query 𝑊𝑄. To support the “near-Laplace tails” and the error–GSR trend more broadly, please: (i) report CCDFs for several representative weight matrices beyond layer-0 𝑊𝑄; (ii) show activation CCDFs at the actual quantization insertion points (e.g., inputs to the main linear projections), not only pre-RMSNorm; and (iii) include the error-vs-𝑟 plots for a few deeper layers to verify the universality of the slope across the stack.

3. The paper would benefit from a side-by-side comparison with DuQuant [1] under matched settings—same checkpoints, KV precision, group sizes, calibration set size, evaluation harness/version, and decoding setup. Because FRTQ emphasizes tiny calibration (e.g., 1×2048 tokens for rotations, 128×2048 for weight calibration) and fused rotations with near-zero runtime overhead, an apples-to-apples table would clarify the accuracy/latency/compute trade-offs relative to DuQuant. If DuQuant requires materially different calibration or runtime transforms, please discuss how these differences affect the headline numbers.

[1] Lin et al. DuQuant: Distributing Outliers via Dual Transformation Makes Stronger Quantized LLMs. NeurIPS 2024.

**Questions:**

Please see the Weakness.

---

> ### Author Response · Authors · 2025-12-02
> **Response to reviewer Un2c**
>
> We sincerely thank the reviewer for the careful reading and for highlighting the simplicity and practicality of our PTQ pipeline, as well as the usefulness of the GSR analysis and ablations. We address the main concerns as follows.
>
> 1.	On Figure 1 and the supporting distribution/error plots.
>
> We agree that the current version of Figure 1 is hard to read. In the camera-ready version we will (i) increase the font size of all labels and tick marks and (ii) provide a higher-resolution version of the figure. In addition, we will extend the empirical evidence for the “near-Laplace tails” and the error–GSR trend by adding more plots in the appendix: CCDFs for multiple representative weight matrices across deeper layers, CCDFs for activations at the actual quantization insertion points (e.g., inputs to the main linear projections), and error-vs-$r$ curves for several deeper layers. This will make the empirical support for the stated trends more comprehensive.
>
> 2.	On comparison with DuQuant and the relationship between DuQuant and FRTQ.
>
> We were not aware of DuQuant at the time of the initial submission and therefore did not include it in our experiments. After reading your review we carefully studied DuQuant and agree that it is a very strong and elegant approach. Unfortunately, due to time constraints we were not able to implement DuQuant and run a matched comparison (same checkpoints, KV precision, group sizes, calibration size, and evaluation harness) before the rebuttal deadline. Conceptually, however, our main goal in this paper is somewhat orthogonal to DuQuant: we focus on designing and analyzing the GSR metric and showing that it can locally guide the design of lightweight, decoupled optimizations (Uniform Preconditioning + DFRot refinement on activations, tSVD + row-wise updates on weights). In this sense, FRTQ is intended as a plugin-style refinement that can be applied on top of existing PTQ schemes, rather than a competing global transformation like DuQuant. Our optimization is fully local and decoupled for each matrix and does not require a global dual transformation across the entire model, whereas DuQuant explicitly performs global dual transformations to redistribute outliers. We believe the two lines are complementary rather than in direct competition: in future work we plan to (i) implement DuQuant in our codebase, (ii) run an apples-to-apples comparison under matched settings, and (iii) explore combining GSR-guided local refinements with DuQuant-style global transformations.

---

### Official Review · Reviewer_Vz2p · 2025-11-01

**Soundness:** 2
**Presentation:** 3
**Contribution:** 2
**Rating:** 4
**Confidence:** 3

**Summary:**

This paper introduces Flattened Rotation tSVD Quantization (FRTQ), a post-training quantization (PTQ) framework designed for ultra-low-bit settings (e.g., W4A4). The key idea is to minimize a novel metric called the Grid-to-Standard-Deviation Ratio (GSR), defined as the quantization grid size Δ divided by the standard deviation σ. The authors theoretically and empirically show that quantization error scales linearly with GSR.

**Strengths:**

- The paper’s motivation is clear — it aims to provide a quantifiable measure (GSR) for characterizing quantization difficulty.
- The authors combine theoretical and empirical analysis to validate the GSR metric and apply it to guide improvements in quantization.
- The proposed approach is experimentally evaluated on multiple models, showing partial evidence of its effectiveness.

**Weaknesses:**

- The authors need to compare with related recent baselines, such as FlatQuant.
- The paper states negligible overhead, yet provides no runtime/memory comparisons. Numbers are needed to substantiate practicality.
- The study focuses on perplexity and a set of zero-shot commonsense tasks; broader benchmarks (e.g., MMLU/BBH) would better probe reasoning/generalization and could reveal trade-offs.
- What's the definition of $R_1$ in Table 1 caption?

**Questions:**

See weaknesses above.

---

> ### Author Response · Authors · 2025-12-02
> **Response to reviewer Vz2p**
>
> We thank the reviewer for the thoughtful comments and helpful suggestions. We respond to the main concerns below.
>
> 	1.	On comparison with FlatQuant and the relationship between FlatQuant and FRTQ.
> At the time of writing the initial submission, we were not yet familiar with FlatQuant and therefore did not include it in our experimental comparison. After reading your review we carefully studied FlatQuant and agree that it is a very appealing and strong method. Due to time constraints we were unfortunately not able to implement it and run a full set of experiments before the rebuttal deadline. Conceptually, however, we view FRTQ as a plugin-style refinement that can be applied on top of existing PTQ algorithms: it operates as an additional GSR-guided “micro-tuning” step on the rotations and tSVD preconditioning, rather than as a competing end-to-end quantization scheme. In the current paper we instantiate FRTQ on top of DFRot, but the same GSR-guided refinement can in principle be combined with FlatQuant as well. In this sense, FlatQuant and FRTQ are orthogonal and not contradictory; we will clarify this positioning in the revised version.
>
> 	2.	On broader benchmarks such as MMLU/BBH.
> We focused our main experiments on perplexity and several zero-shot commonsense tasks because the prior PTQ works we compare against (e.g., QuaRot/DFRot) did not report results on MMLU/BBH, making a systematic and fair comparison across methods less straightforward. Following your suggestion, we ran an additional experiment on LLaMA-3-8B on MMLU under the W4A4 setting. We obtained:
> 	•	Hadamard baseline: 54.33
> 	•	DFRot: 56.24
> 	•	FRTQ (ours): 56.27
> The results show that FRTQ tracks DFRot very closely on this benchmark, and the extra GSR-guided refinement leads to only a small additional gain in this particular setting. We will report these numbers and explicitly acknowledge the current lack of broader reasoning benchmarks (MMLU/BBH) as a limitation and an important direction for future work.
>
> 	3.	On runtime, latency, and optimization cost.
> During inference, FRTQ follows the widely used dequantize-then-matmul pattern (weights are stored in low precision, dequantized back to floating point, and then multiplied), which is also adopted by many existing W4A4 systems. The only additional cost compared to a plain W4A4 baseline comes from the tSVD-based factorization of the weight matrices: in our implementation this leads to roughly $\approx 10\%$ increase in end-to-end inference time, while latency and throughput are otherwise the same as running the underlying quantized LLM with the same kernels. The offline optimization procedure is also lightweight: each matrix is optimized completely independently, and on an H20 GPU with parallel processing the entire optimization for an 8B model finishes in about 10 minutes. We will clarify these points and add a concise latency/memory table in the revised version to substantiate the practicality claim.
>
> 	4.	On the definition of $R_1$ in Table 1.
> We apologize for the missing definition in the caption. The matrix $R_1$ in Table 1 is exactly the $R_1$ rotation matrix from QuaRot, i.e., the orthogonal matrix that is multiplied before/after each RMSNorm (the same $R_1$ that is fused into the RMSNorm-associated linear layers in QuaRot/DFRot). We will correct the caption and ensure that this notation is clearly defined and consistent across the paper.

---

### Meta-Review · Area_Chair_LoSV · 2026-01-08

**Summary:**

This paper introduces FRTQ (Flattened Rotation tSVD Quantization), a post-training quantization framework for ultra-low-bit (W4A4) quantization of large language models. The key contribution is the introduction of the Grid-to-Standard-Deviation Ratio (GSR) as a unified metric to guide quantization decisions. The method combines Uniform Preconditioning (UP) with refined rotation for activations and truncated-SVD (tSVD) with row-wise updates for weights. The paper demonstrates competitive results across LLaMA model families, approaching QAT-level performance without gradient-based training.

All four reviewers found merit in the core ideas - particularly the GSR metric and the principled approach to handling massive activations. However, reviewers raised several important concerns regarding experimental comparisons, presentation clarity, and breadth of evaluation. The authors provided responsive rebuttals addressing most concerns, though time constraints prevented them from implementing additional baselines like FlatQuant and DuQuant.

While this paper presents interesting ideas, particularly the GSR metric and the principled combination of rotation and low-rank methods, it falls short of the acceptance bar due to:

1.	Incomplete experimental evaluation: The absence of comparisons with FlatQuant (mentioned by three reviewers) and DuQuant represents a critical gap, especially given that authors position FRTQ as potentially complementary to these methods. The limited model family coverage (LLaMA only) and benchmark scope further weaken the empirical contribution.

2.	Presentation quality issues: Multiple reviewers found the paper difficult to follow, with confusing notation, hard-to-read figures, and algorithm details relegated to appendices. These issues significantly impact the paper's accessibility and reproducibility.

3.	Incremental contribution relative to DFRot: While GSR is a useful metric, the empirical gains over DFRot are relatively modest (often <1% on zero-shot tasks). The relationship to and advantages over concurrent work like FlatQuant and DuQuant remain unclear.

The core ideas have merit and could potentially form the basis of a strong paper with:

•	Comprehensive comparison with FlatQuant and DuQuant under matched conditions.

•	Broader evaluation across model families and challenging reasoning benchmarks.

•	Significantly improved presentation with clearer notation and main-text algorithm descriptions.

•	Deeper analysis of when FRTQ provides advantages over existing methods.

**Reviewer Concerns:**

Addressed by Rebuttal:

•	Runtime/memory overhead clarification: Authors clarified ~10% overhead and provided details on optimization cost (~10 minutes for 8B model).

•	Definition of R1 notation: Acknowledged and will be corrected in revision.

•	Laplace distribution assumption: Authors explained GSR validity extends beyond Laplace to symmetric, finite-variance distributions.

•	Model family coverage: Authors acknowledged limitation and committed to broader evaluation in future work.


Partially Addressed:

•	MMLU/BBH benchmarks: Authors ran additional MMLU experiments on LLaMA-3-8B showing FRTQ (56.27) closely tracks DFRot (56.24), but acknowledged need for more comprehensive reasoning benchmarks.

•	SpinQuant/OSTQuant comparison: Authors clarified terminology confusion (PTQ vs. QAT) and positioning of FRTQ as a lightweight alternative, though performance gaps remain in some cases.


Outstanding Concerns:

•	Missing key baselines (Critical): All reviewers noted absence of FlatQuant, and Reviewer Un2c specifically requested DuQuant comparison. Authors acknowledged these are important but could not implement due to time constraints. This is a significant gap as FlatQuant is highly relevant recent work.

•	Presentation issues (Moderate):

a.	Reviewer LRtm rated presentation as "poor" citing confusing notation and difficulty following the method description.

b.	Multiple reviewers noted Figure 1 text is too small.

c.	Algorithm description relegated to appendix makes main text hard to follow.

•	Limited evaluation scope (Moderate):

a.	Experiments limited to LLaMA family; no encoder-only, encoder-decoder, or vision-language models.

b.	Focused primarily on perplexity and zero-shot tasks; limited reasoning benchmarks.

c.	No comparison with mixed-precision methods beyond those included.

**Reviewer Scores:**

Reviewer Vz2p (Initial: 4): Likely to remain at 4. While rebuttal addressed some concerns (runtime clarification, R1 definition), the missing FlatQuant comparison and limited broader benchmarks remain unresolved.

Reviewer Un2c (Initial: 4): Likely to remain at 4. The rebuttal was comprehensive on technical details (Laplace assumption, figure improvements), but the absence of DuQuant comparison and limited scope keep concerns alive.

Reviewer LRtm (Initial: 4): Likely to remain at 4. The presentation concerns were acknowledged but not fully resolved in rebuttal, and key baseline comparisons are missing. The clarification on SpinQuant/OSTQuant helps but doesn't fully address performance gaps.

Reviewer eAW1 (Initial: 6): Likely to remain at 6. This reviewer was most positive, praising innovation and practicality. Rebuttal addressed main theoretical and runtime concerns adequately. The reviewer explicitly stated "would not mind if paper is rejected," indicating borderline confidence.

---

### Decision · Program_Chairs · 2026-01-26

Reject